# *Microplitis manilae* Ashmead (Hymenoptera: Braconidae): Biology, Systematics, and Response to Climate Change through Ecological Niche Modelling

**DOI:** 10.3390/insects14040338

**Published:** 2023-03-30

**Authors:** Mostafa Ghafouri Moghaddam, Buntika A. Butcher

**Affiliations:** Integrative Ecology Laboratory, Department of Biology, Faculty of Science, Chulalongkorn University, Phayathai Road, Pathumwan, Bangkok 10330, Thailand; mostafa.g@chula.ac.th

**Keywords:** armyworms, biological control, climate change, environmental suitability, Microgastrinae, parasitoid wasp

## Abstract

**Simple Summary:**

This study focused on the parasitoid wasp *Microplitis manilae*, which is an important natural enemy of noctuid caterpillars, including the pest species of armyworms. The parasitoid wasp is here redescribed and illustrated based on the holotype, and an updated list of all *Microplitis* species attacking *Spodoptera* spp. is provided along with a discussion of host-parasitoid-food plant associations. We used a maximum entropy model and a quantum geographic information system to simulate the distribution of *M. manilae* in present and future periods under four greenhouse gas concentration scenarios. The results indicated that the suitable habitats for *M. manilae* are mainly in tropical and subtropical countries, and these are expected to expand in the future due to climate change. The study offers a basis for environmental protection and pest management.

**Abstract:**

The parasitoid wasp *Microplitis manilae* Ashmead (Braconidae: Microgastrinae) is an important natural enemy of caterpillars and of a range of noctuids, including pest species of armyworms (*Spodoptera* spp.). Here, the wasp is redescribed and, for the first time, illustrated based on the holotype. An updated list of all the *Microplitis* species attacking the noctuid *Spodoptera* spp. along with a discussion on host-parasitoid-food plant associations is offered. Based on information about the actual distribution of *M. manilae* and a set of bioclimatic variables, the maximum entropy (MaxEnt) niche model and the quantum geographic information system (QGIS) were explored to predict the potential distribution of this wasp in a global context. The worldwide geographical distribution of potential climatic suitability of *M. manilae* at present and in three different periods in the future was simulated. The relative percent contribution score of environmental factors and the Jackknife test were combined to identify dominant bioclimatic variables and their appropriate values influencing the potential distribution of *M. manilae*. The results showed that under current climate conditions, the prediction of the maximum entropy model highly matches the actual distribution, and that the obtained value of simulation accuracy was very high. Likewise, the distribution of *M. manilae* was mainly affected by five bioclimatic variables, listed in order of importance as follows: precipitation during the wettest month (BIO13), annual precipitation (BIO12), annual mean temperature (BIO1), temperature seasonality (BIO4), and mean temperature during the warmest quarter (BIO10). In a global context, the suitable habitat of *M. manilae* would be mainly in tropical and subtropical countries. Furthermore, under the four greenhouse gas concentration scenarios (representative concentration pathways: RCP2.6, RCP4.5, RCP6.0, and RCP8.5) in the future period of the 2070s, the areas with high, medium, and low suitability showed varying degrees of change from current conditions and are expected to expand in the future. This work provides theoretical backing for studies associated with the safeguarding of the environment and pest management.

## 1. Introduction

Biological control agents represent a sustainable pest management option that helps to maintain pest populations under accepted levels [1]. Parasitoid wasps from the Braconidae family represent a very important group of biological control agents as they have a wide range of insect hosts with different degrees of specialization [2]. Microgastrinae is a highly specialized Braconidae subfamily that exclusively attacks caterpillars [3]. Microgastrines form one of the most diverse groups of parasitoid wasps worldwide distributed, a characteristic that makes this group an ideal candidate for comparisons between populations in different locations [2,3]. This will allow researchers to understand how an organism may adapt to different environments, and how different populations may be genetically distinct. *Microplitis* Foerster, 1863 is one of the mega-genera within the microgastrine group, is considered an early diverging taxon, and is well represented in all biogeographic regions. Species are endoparasitoids with both solitary and gregarious larval development, and their hosts are mainly macrolepidopterans (e.g., Erebidae, Geometridae, Lasiocampidae, Noctuidae, Nymphalidae, Papilionidae, Pieridae, Saturniidae, etc.) [4]. Within the Noctuidae family, coined the ‘pest clade’ [5], the genus *Spodoptera* Guenée, known as the armyworm, is one of the major pests that causes great damage in terms of monetary loss to agriculture worldwide [6]. The genus is native to the tropical regions of the Western Hemisphere, from the United States to Argentina [7].

There are some scant published records of *Microplitis* attacking *Spodoptera* larvae. To date, only 15 *Microplitis* species have been reported parasitizing *Spodoptera* species [8]. Elucidating the diversity of *Microplitis* species that use this pest-rich genus as hosts is a task that is not currently being undertaken in any of the six biogeographical regions, and consequently, the host associations remain largely unknown.

*Microplitis manilae* Ashmead, 1904 is a species reported in the Australasian, Oriental, and Palaeartic Regions. In Asia, it is widely distributed in Southeast Asia, including Indonesia, Malaysia, the Philippines, Thailand, and Vietnam [3]. The wasp has already been part of integrated pest management programs for the prevention and control of populations of *Spodoptera* larvae with remarkable results in China [9,10]. This species has also been introduced to the United States (Florida) from Thailand to control the fall armyworm, *Spodoptera frugiperda* (J. E. Smith), but has never been established [11]. Temperature and precipitation, along with host distribution, are key factors that ultimately shape the occurrence of *M. manilae*. These parameters affect the establishment of *M. manilae*, as well as how it will spread and potentially thrive in a given environment [12].

Ecological niche models (ENMs) belong to a category of techniques that infer the relationship between species distribution and bioclimatic factors. This method involves establishing a connection between the occurrence data of the target species and the bioclimatic variables present in the corresponding locations. By leveraging this relationship, it becomes possible to estimate the distribution of regions that meet the niche prerequisites of the target species, subsequently identifying them as parts of the potential distribution [13]. ENMs serve as indispensable tools in ecological research [14]. Over the past few decades, ENMs have extensively gained significant traction in studying the distribution of species. Among available ENM techniques, the maximum entropy model (MaxEnt) has become one of the most popular tools for modelling species distribution. Numerous studies have illustrated the benefits of utilizing this simulation software for precise predictions, especially when the species being studied has a restricted distribution dataset and only necessitates a small sample size [13,14,15,16]. Since its launch, the MaxEnt model has been commonly applied in assessing the potential distribution of species, the risk of species invasion, pest and disease spread and control, and the protection of endangered plants and animals [13,14,15]. Additionally, the MaxEnt model has also been utilized to predict the suitable areas for parasitoid wasps given present and future climate conditions. This approach assesses how climate change impacts the distribution of parasitoid wasps, thereby establishing a foundation for further research on these organisms [16]. The quantum geographic information system (QGIS) is a free and open-source cross-platform information system that allows users to create, edit, visualize, analyze, and publish geospatial information.

In this study, *Microplitis manilae* is redescribed and, for the first time, illustrated based on the holotype. Additionally, the host-parasitoid-food plant associations data of all *Microplitis* species attacking *Spodoptera* hosts are presented. Finally, the MaxEnt and QGIS technologies were used to analyze the environmental suitability of *M. manilae* by combining known distribution and environmental data in a global context. Predicting the current and future potential distribution of *M. manilae* will provide a theoretical basis for pest control, particularly for armyworms.

## 2. Materials and Methods

### 2.1. Sampling, Preparation, Terminology, and Photography

The Thai specimens of caterpillar and parasitoid wasp examined in this paper were collected and reared by the research team of the Integrative Ecology Laboratory (IE-Lab) (Department of Biology, Chulalongkorn University, Bangkok, Thailand) from Lainan, Wiang district (Nan Province, Thailand). The morphological terminology and measurements followed those of Fernandez-Triana et al. [17] and Ghafouri Moghaddam et al. [18]. Preparation of specimens, labelling, comparative measurements for ratios, and preparation of photographic plates followed the methods described by Ghafouri Moghaddam et al. [18,19,20]. Geo-referenced data for *Microplitis manilae* were obtained from Yu et al. [21] and Fernandez-Triana et al. [3]. A distribution map was created using the SimpleMappr [22] and processed with Adobe Photoshop^®^ CC 2022 software (Adobe Co., San José, CA, USA). Photographs of Thai specimens were taken using a Leica^®^ M205 C microscope with montage multi-focus, interactive measurement, and fusion optics stereomicroscope combined with Leica Application Suite. All images were processed using Adobe Photoshop^®^ CC ver. 23.3 2022 software, and the same software was used to create the figure plates. Throughout the text, the acronyms T1, T2, T3, and so on are used for mediotergites 1, 2, 3, and so on.

### 2.2. Rearing Thai Specimens of Caterpillars and Parasitoid Wasp

Several parasitized caterpillars of the family Noctuidae were collected during 2012–2013 and fed on three varieties of *Brassica oleracea* L., Brassicaceae: broccoli (var. *italica*), cabbage (var. *capitata*), and cauliflower (var. *botrytis*). The cruciferous vegetables were cultivated on a cabbage farm located in upper Northern Thailand (the Nan province). There were three lines (LN1–3) in each variety plot. This was to ensure that the comparison of the different varieties was conducted in a controlled and systematic manner. Three lines from each variety plot were sampled. The caterpillars were collected from the same farm in their natural habitat and put in a dry and clean plastic container. The container was sealed with a stretch film wrap to maintain sterility and then transported to the IE-Lab. Leaves of the three varieties of *B*. *oleracea* were collected to feed the caterpillars. The caterpillars were kept at room temperature 24 ± 1 °C, 55 ± 5% RH, and 12:12 h L:D until either the emergence of the lepidopteran or the parasitoid occurred. The containers were monitored daily and the number of cocoons, adult emergence, and the number of males and females were recorded.

### 2.3. Species Identifications

The species recognition of the parasitoid was made via both morphological and biological evidence following the *Microplitis* revision by Austin and Dangerfield [23]. Additionally, species identification keys and original description were also consulted [8,23,24]. *Microplitis* species concepts for those attacking *Spodoptera* hosts are based on Gupta [8].

### 2.4. Ecological Niche Modelling

The key prerequisite for constructing a niche model is the adequate availability of existing species records [16]. The records of the geographical distribution of *M. manilae* were obtained by consulting different sources, including relevant literature and books and the Global Biodiversity Information Facility (GBIF, https://www.gbif.org/; accessed on 7 February 2023), and integrating with GPS field survey data. All latitude and longitude coordinates obtained were homogenized using the World Geodetic System 1984 (WGS84), and distribution points were confirmed by Google Earth^®^ Pro ver. 7.3.6.9345 (https://earth.google.com/; accessed on 7 February 2023). Distribution points were imported into QGIS^®^ software. The buffer analysis method was applied to filter out the acquired distribution points to eliminate the impact of over-fitting simulation arising from large spatial correlation [25]. Given that the bioclimatic variables had a spatial resolution of 2.5 arcminutes (about 4.5 km^2^), the buffer radius was set to 1.5 km. When the distance between the distribution points was less than 3 km, only a single point was retained. Ultimately, a total of 29 valid sites were acquired that met the acceptance criteria and were exported as a CSV (comma-separated values) file for further model analysis to provide valuable insights.

This study extracted bioclimatic variables from the Worldclim database to comprehensively investigate the impact of climate on the spread of *M. manilae* on a global scale (Version 2.1, http://www.worldclim.org/; accessed on 14 February 2023). The bioclimatic variables used in the model include 19 bioclimate variables and the elevation data selected as a topographic factor (Table 1). These bioclimatic variables were used for erecting the current scenario and the data corresponding to the average values compiled in the period between 1970 and 2000, and the corresponding raster files (images built from pixels) were downloaded at a spatial resolution of 2.5 arcminutes. The future climate data for the 2070s were obtained from the Climate Change, Agriculture and Food Security (CCAFS) website. The Fifth Assessment Report (AR5) of the Intergovernmental Panel on Climate Change (IPCC) considered four greenhouse gas concentration scenarios [26]. In this study, three distinct representative concentration pathways (RCPs) were selected to model the distribution of the species, including the minimum greenhouse gas emission scenario (RCP2.6), a medium greenhouse gas emission scenario (RCP4.5), and the maximum greenhouse gas emission scenario (RCP8.5).

The potential distribution of *M. manilae* was simulated using MaxEnt™ ver. 3.4.4 software, which combined the screened climatic factors with the determined distribution points. The repetition training number was set to 10 to mitigate the uncertainty caused by abnormal values in the bioclimatic variables associated with the randomly chosen training points. The maximum number of background points was set to 10,000, and bootstrap was selected as the replicated run type. The default settings for the remaining MaxEnt parameters were retained. The Jackknife test of the MaxEnt software was utilized to determine the percent contribution of each bioclimatic variable during the construction of the initial model.

The distribution territory of *M. manilae* in a global context was extracted by QGIS, and the climatic suitability for the species was analyzed. The output of the MaxEnt software simulation varied between 0 to 1. This probability surface represents the likelihood of suitable habitat for the species being modelled across the study area. Higher outputs closer to 1 from the MaxEnt software simulation indicate a greater likelihood of the species being present. Given the current distribution of *M. manilae* and all the information contained in the IPCC report regarding possible future changes in anthropogenic greenhouse gas emissions [27], the habitat suitability was divided into four levels according to the probability and each one was represented by different colors: high-suitability area, medium-suitability area, low-suitability area, and unsuitable area. MaxEnt uses the logistic output format, which means that the probability surface is divided into 10 equally spaced bins. Each bin represents a range of probabilities, and the number of occurrences and background points falling into each bin is used to calculate the suitability values. Therefore, the levels of suitability are not divided equally within the 0–1 range by default. In the present study, the default parameters of the MaxEnt software were regularization multiplier (RM) = 1, feature classes (FC) = LQHPT (L (linear), Q (quadratic), H (hinge), P (product), and T (threshold)).

AUC_DIFF_ (the difference between the training set AUC (area under the curve) and test set AUC) and the omission rate were used to test the fit of the model to species distribution. We used a similarity index, the multivariate environmental similarity surfaces (MESS), with the default settings of MaxEnt to identify areas that did not meet a certain threshold of similarity to the training data and exclude them from the analysis. This default value is usually set to two standard deviations away from the mean similarity value of the training data. Any area with a similarity value below this threshold was considered dissimilar to the training data and excluded from the analysis. The use of MESS to exclude dissimilar areas can help to improve the accuracy and ecological relevance of the species distribution model. By excluding areas that are significantly different from the training data, the model can focus on areas that are more likely to support the target species. The higher accuracy of the constructed model can be inferred when the test omission rate is closer to the theoretical omission rate [28]. The Akaike information criterion (AIC) was used to evaluate the fitting degree and complexity of different parameter combinations. Removing highly correlated variables is a common practice in species distribution modelling because it can improve the accuracy and interpretability of the model. Following the optimization, solely the optimal parameters were used for simulating and predicting the suitable habitat of *M. manilae* across different periods. The precision of the simulation outcomes was assessed through the receiver operating characteristic curve (ROC), and the area under the curve (AUC) was used to evaluate the predictive performance of the model [29]. The value of AUC ranges between 0 and 1, with 1 representing a perfect prediction. Values greater than 0.9 are considered to indicate a high level [30].

### 2.5. Depositories

The specimens treated in this study were deposited in the Collection of the Insect Museum, Chulalongkorn University Museum of Natural History, Bangkok, Thailand (CUMZ) and the National Museum of Natural History, Smithsonian Institution, Washington, DC, USA (USNM).

## 3. Results

### 3.1. Taxonomic Account

***Microplitis manilae*** **Ashmead, 1904**

*Microplitis manilae* Ashmead, 1904.

*Snellenius manilae* (Ashmead, 1904).

*Microgaster manilae* (Ashmead, 1904).

(Figure 1, Figure 2 and Figure 3)

**Type examined. HOLOTYPE**, **female** PHILIPPINES, Manila PI, no date, W. A. Stanton leg., type No. 7715, QR barcode: USNMENT 00809863 (USNM).

**Additional material examined. LN1**: one female and one male • THAILAND, Nan, Wiang Sa, Lainan, 12.x.2012, reared on *Spodoptera litura* larvae, feeding on cauliflower, K. Chansri leg. (CUMZ); one female and one male • same data except for 13.x.2012; one female • same data except for 13.iii.2013, feeding on cabbage; one female • same data except for: 13.x.2013, feeding on broccoli; one female and one male • same data except for 12.x.2012, feeding on broccoli; **LN2**: one female • THAILAND, Nan, Wiang Sa, Lainan, 21.ix.2012, reared on *Spodoptera litura* larvae, feeding on broccoli, K. Chansri leg. (CUMZ); one male • same data except for 15.xii.2012; one female • same data except for 17.xi.2012; one male • same data except for 17.xi.2012, feeding on cabbage; one female • same data except for 23.ix.2012, feeding on cabbage; three females and two males • same data except for 11.i.2013, feeding on cauliflower; two females • same data except 13.iii.2013, feeding on cabbage; one female • same data except for 13.i.2013, feeding on cauliflower; **LN3**: one female and two males • THAILAND, Nan, Wiang Sa, Lainan, 17.xii.2012, reared on *S. litura* larvae, feeding on broccoli, K. Chansri leg. (CUMZ); two females and one male • same data except for 16.xii.2012; one female • same data except for 12.i.2013; one female and one male • same data except for 16.xii.2012, feeding on cauliflower; one female • same data except for 18.xi.2012, feeding on cabbage.

**Diagnosis.** Wings infuscated, pterostigma uniformly dark brown; mesoscutum finely reticulate-punctate, often smoother on lateral lobes; medial furrow slightly impressed, crenulate-punctate, weakly-to-moderately well-defined; scutellar scutoscutellar with seven distinct carinae; dorsal scutellum faintly punctate medially; T1 about 2.0 × as long as wide, parallel sided, slightly narrowed at apex with smooth apical swelling, finely rugose punctate in posterior half and laterally except apical patch; median length of T3 about 1.4 × more than T2; hind tibia with median one third pale testaceous to white, apical one third with black infuscation.

**Redescription.** Body length 2.7 mm, fore wing length 2.3 mm and antenna length 2.9 mm.

*Color.* Head and mesosoma black; antenna, coxae, metafemur, and metatarsus dark brown; palps brown to yellow whitish; wings infuscated, venation brown, pterostigma uniformly dark brown; pro- and mesofemur, pro- and mesotibia brown; pro- and mesotarsus brown to dark brown; metatibia with basal or medial half to one third pale testaceous to white, apical one third with black infuscation; metabasitarsus with pale extreme base; T1 dark brown, laterotergites yellowish brown; T2 with dark brown median field indicated by indistinct oblique grooves, laterotergites yellowish brown; T3 with mixture of brown and dark brown patches; remaining tergites and sternites dark brown.

*Head* (Figure 2C–E). Circular in frontal view, head maximum height/head maximum width: 0.8, head maximum height/temple maximum length: 3.5, head maximum height/eye maximum length: 1.5; lateral temples hidden behind eyes in frontal view, in lateral view width of temple equal to width of eye; face and clypeus finely reticulate-punctate, densely setose, clypeal maximum width/clypeal maximum length: 1.9, clypeal maximum width/ocular-mandibular line: 2.0; face slightly convex, with faint medial, longitudinal carina in dorsally, width of face (at widest) 0.5 × width of head; inner margins of eyes straight to slightly emarginate near to antennal sockets; upper frons, vertex and temples finely punctate, with short sparse setae, upper frons with small pit below median ocellus; lower frons smooth and shining, with a few transverse striae above antennal sockets curving around sockets laterally; occiput moderately excavate, smooth except for fine punctation laterally which extends onto temples; eyes setose, eye maximum length/eye maximum width: 2.1, eye maximum length/temple maximum length: 2.5, eye maximum length/ocular-mandibular line: 3.5, inter-ocular line/ocular-mandibular line: 3.1; ocelli forming an obtuse triangle, posterior ocellar line/ocellus diameter: 2.6, oculo-ocellar line/ocellus diameter: 2.2, posterior ocellar line/oculo-ocellar line: 1.1; antenna moderately robust, longer than body, body length/antenna length: 0.9, first antennal flagellomeres length/first antennal flagellomeres width: 3.2, second antennal flagellomeres length/second antennal flagellomeres width: 2.6, antennal flagellomeres length 14/antennal flagellomeres width 14: 2.3, antennal flagellomeres length 15/antennal flagellomeres width 15: 2.5, antennal flagellomeres length 16/antennal flagellomeres width 16: 3.2.

*Mesosoma* (Figure 1C,D and Figure 2F,G,I). Mesoscutum finely reticulate-punctate; notauli visible and slightly impressed, crenulate, meeting posteriorly to form broad coarser longitudinally reticulate-punctate area, medial furrow slightly impressed, crenulate-punctate, weakly-to-moderately well-defined; scutoscutellar sulcus broad, deep, divided by seven carinae; area anterior and lateral to scutoscutellar sulcus smooth; dorsal scutellum convex, setose, faintly punctate medially, postero-lateral margin bordered by crenulate furrow, petering out to a few punctures anteriorly, posterior part slightly upturned, this area coextensive with rugulose medial posterior band; maximum height of lateral face of mesoscutellum height/maximum height of mesoscutellum lunula height: 4.6; propodeum with median longitudinal carina surrounded by coarse rugosity, transverse carina absent, posterior half of propodeum/anterior half of propodeum: 1.7; lateral pronotum with deep oblique crenulate furrow, becoming more reticulate in posterior part, areas ventral and dorsal to furrow finely reticulate-punctate; mesopleuron mostly smooth and shining, dorsal epicnemial area and anterior margin finely punctate, with dense long fine setae; epicnemial furrow deep, crenulate, reaching anteriorly to margin of mesopleuron; precoxal groove crenulate to crenulate-punctate, upturned to meet epicnemial furrow; mesosternum finely punctate, setose.

*Wings* (Figure 1B and Figure 2B). Fore wing 2.7 × as long as wide, pterostigma broad, pterostigma maximum length/pterostigma maximum width: 2.3, (RS + M)b 0.9 × 2M, 2RS 1.3 × r, R1 0.6 × distance from pterostigma to 3RSb, 1M very slightly curved, areolet quadrangular, 1CUb 2.8 × as long as 2CUa, first submarginal cell 1.6 × as long as wide; hind wing with 1M slightly sinuate, R vein present.

*Legs* (Figure 1D,E). Metacoxa smooth, setose, metacoxa maximum length/metacoxa maximum width: 1.9; metafemur length/metafemur width: 4.4; metatibia length/metabasitarsus length: 2.5; metabasitarsus length/inner metatibial spur length: 2.6, ratio of lengths of metatarsus segments 1–5 3.4:1.3:1.0:0.7:1.2; tarsal claws small, simple.

*Metasoma* (Figure 1C and Figure 2H,J). T1 maximum length/T1 maximum width at anterior margin: 2.1, T1 maximum length/T1 maximum width at posterior margin: 2.9, T1 maximum width at anterior margin/T1 maximum width at posterior margin: 1.3, finely rugose-punctate in posterior half, smooth basally, parallel-sided except for slight constriction medially and slight narrowing apically, apical surface with smooth convex swelling; T2 maximum width at posterior margin/T2 maximum length: 2.4, smooth, with shield-shaped median field indicated by oblique grooves; suture between T2 and T3 reduced to slight depression; median length of T3 about 1.4 × more than T2; T3–T7 each with one or two transverse rows of setae posteriorly, becoming denser laterally; hypopygium smooth, sparsely setose; ovipositor sheaths slightly curved, with sparse moderately long setae and tuft of fused setae apically.

**Male** (Figure 3)**.** As for female, except as follows: head slightly less sculptured; mesoscutum more sparsely punctate; notauli and medial furrow with weak sculptures and slightly crenulate; 1M sometimes slightly more curved; margins of T1 slightly more emarginate. However, they differ in some measurements (second antennal flagellomere length/second antennal flagellomere width, antennal flagellomeres length 16/antennal flagellomeres width 16, metafemur length/width, first segment of metatarsus length), and coloration, such as scape (black to brown yellowish), flagellomeres (black), fore and mid legs (dark brown to yellow brown) T1-T2 (dark brown).

**Distribution** (Figure 4)**. Australasian:** Australia (Queensland), Papua New Guinea; **Oriental:** China (Guangdong, Taiwan, Zhejiang), India, Indonesia, Malaysia, Philippines, Japan (Ryukyu Islands), Thailand, Vietnam; **Palaearctic:** South Korea.

**Host** (Table 2)**.** This table contains a comprehensive list of the host associations that were studied as part of the research, as well as additional details such as the host plant, solitary/gregariousness of wasp larvae, cocoon color, and the degree of species support by morphological, molecular, and biological data. The collected cocoon was solitary, oval, and light brown. The cocoon remained attached to the underside of the leaf with light brown cotton fibers.

**Notes.** The published host record from the bean pod borer, *Maruca vitrata* (F.) [Crambidae, Pyraustinae] reported from the Philippines [31] is probably incorrect because such microlepidopteran moth caterpillars are unlikely to be attacked by *Microplitis* species. In many older classifications, the Crambidae were included in the Pyralidae as a subfamily.

### 3.2. Ecological Niche Modelling

Nine out of the nineteen key bioclimatic variables (Table 1) were screened out according to the percent contribution, and then the species distribution model of *Microplitis manilae* was reconstructed. The percent contribution and permutation importance of those nine variables are listed in order of importance: precipitation during wettest month (BIO13), precipitation during coldest quarter (BIO19), mean diurnal range (BIO2), mean temperature during warmest quarter (BIO10), elevation (ELV), temperature seasonality (BIO4), annual mean temperature (BIO1), precipitation seasonality (BIO15), and annual precipitation (BIO12) (Table 3). According to the results of the jackknife test, the accumulated percent contribution of the first three bioclimatic variables (67.4%, 18.6%, and 5.3%, respectively) accounted for more than 90%. Consequently, the nine bioclimatic variables were found to contain effective information regarding the suitable habitat of *M. manilae* and were deemed crucial in the simulation of its potential geographical distribution. Elevation is a static feature and does not change over time, and the elevation layer does not provide any additional information; thus, neither were considered as input in predicting future scenarios. The ROC curve generated by the model demonstrated an AUC value of 0.925, which signified a robust level of predictive accuracy for the model (Figure 5C). The prediction omission rate exhibited strong concurrence with the omission rate of the test sample, suggesting that the model had a favorable predictive performance (Figure 5D). This model was reliable for confirming the potential distribution of *M. manilae* in a global context.

The projection of the suitability distribution for *M. manilae* on a worldwide scale using the optimized MaxEnt model (Figure 5A) showed that the contemporary suitability areas were distributed in tropical and subtropical regions in Asia, mainly in plain areas of low elevation. These areas are located especially in Cambodia, Vietnam, Thailand (central), China (south and east), India (coastal plains in the east and the south, some parts in the north and central; northeastern), Myanmar (south), Philippine (north), Australia (northeast coast plains), and Bangladesh (south). The distribution of the currently suitable area is strongly correlated with the area that is highly suitable, due to the wider distribution range. The moderately suitable areas have the potential to transform into high-suitability areas over the years.

As for habitats predicted under the four greenhouse gas concentrations (RCP2.6, RCP4.5, RCP6.0, and RCP8.5) in the decade of the 2070s (Figure 6A and Figure 7), there were obvious changes. The tendency of all the predicted suitability areas (high, medium, and low) showed a trend of expansion toward other tropical countries located in Africa (Madagascar, Mozambique, Tanzania, Gabon, Equatorial Guinea, Ghana, Ivory Coast), North America (Mexico), Central America (Guatemala, Nicaragua), South America (Ecuador, Guyana, Suriname, Western and Southern Brazil), and even in Western Asia (some small areas, e.g., Iran). Based on model projections for all possible future scenarios, it is anticipated that the area with a high suitability rating will have undergone the most significant expansion by the 2070s, in comparison to the current conditions. As shown in Table 4, under the RCP8.5 scenario, model projections suggest that by the 2070s, the high-suitability areas will expand the most compared to current conditions, accounting for 22.3% of the current predicted ones. Under the RCP4.5 and RCP6.0 scenarios, the extent of the changes will convert low-suitability areas into medium- and high-suitability by the 2070s. The extent of the highly suitable areas will rise under the RCP4.5 and RCP6.0 scenarios, accounting for 18.78% and 18.76% of the current predicted ones, respectively. Under the RCP2.6 scenario, the highly suitable areas will have slightly increased by the 2070s, accounting for 15.71% of the current predicted ones. Between now and the 2070s, there is projected to be a discernible shift in the distribution of areas that are currently deemed highly suitable towards those that are classified as moderately suitable. Alternatively, it is conceivable that by the 2070s, the extent of the changes may have led to the conversion of low-suitability areas into medium- and high-suitability ones (Figure 6 and Figure 7).

All the bioclimatic variables (predictors) affected the potential distribution of *M. manilae*. Precipitation during the wettest month (BIO13) was the most important bioclimatic variable when used alone, corresponding to the long blue band (Figure 5B).

Drawing upon the findings of the IPCC report [27], the range of bioclimatic variables suitable for the distribution of *M. manilae* was demarcated using 0.46 as a threshold. The predicted suitability varied as the values of the chosen bioclimatic variables increased (Figure 8). When the bioclimatic variable value was less than the optimal value, the distribution probability was increased with a rise in the bioclimatic variable value and vice versa. The appropriate range for all nine bioclimatic variables to the potential distribution of *M. manilae* was determined (Table 5). Concerning the five most important bioclimatic variables, the appropriate value range of precipitation during the wettest month (BIO13) was 269.2–763.4 mm, and the optimal value was 435.8 mm. This relatively wide range of values for BIO13 suggests that *M. manilae* can occur under simultaneously humid and warm weather conditions. At 269.2–435.8 mm, the predicted suitability of *M. manilae* increased rapidly with the rises in the precipitation during the wettest month and decreased slowly with increases in the precipitation during the wettest month at 435.8–763.4 mm (Figure 8A). The annual precipitation (BIO12) was 1428.1–3982.5 mm, the predicted suitability of *M. manilae* was higher than 0.46, and the predicted suitability was the highest at 1876.3 mm, reaching 0.69. The medium range of appropriate values for BIO12 suggested that *M. manilae* is highly sensitive to extreme precipitation changes (Figure 8B). The annual mean temperature (BIO1) was lower than 197.3 °C, and the suitability of predicted *M. manilae* was lower than 0.46. With an increase in annual mean temperature, the suitability of prediction increased quickly and reached its peak at 246.7 °C. When the precipitation exceeded 286.4 °C, the suitability of the predicted dropped again below 0.46 (Figure 8C). When the temperature seasonality (BIO4) was 132–4855.6, the predicted suitability of *M. manilae* was higher than 0.46, and the predicted suitability was the highest at 2240.7, reaching 0.62. The medium range of appropriate values for BIO4 suggested that *M. manilae* is susceptible to extreme temperature fluctuation in each of the seasons throughout the year (Figure 8D). A slight change in the mean temperature of the warmest quarter (BIO10) can have a significant effect on the distribution of *M. manilae*, suggesting that the wasps prefer areas with more temperature variation. The appropriate range of the response curve for the mean temperature of the warmest quarter (BIO10) was 198.4–326.2 °C, and the most appropriate value was 292.4 °C (Figure 8E).

## 4. Discussion

### 4.1. Taxonomic Notes

By examining a consistent set of external morphological characters, taxonomists can identify the *Microplitis* species even when the specimens show intraspecific variations in size and/or color [32]. These characters include (a) the sculpturing pattern on the mesoscutum, (b) the shape and sculpturing of the notauli and medial furrow, (c) the sculpturing pattern of the mesoscutellum, (d) the color of the hind legs, (e) the general shape and sculpturing of TI, and (f) the presence of a small median field on T2. Thai *Microplitis manilae* specimens differ slightly from the holotype specimen. These discrepancies are considered intraspecific variations. These differences are mainly in the amount of setae on the mesosoma (dense in Thai specimens vs. somewhat scarce in the holotype) and the coloring pattern of tergites (dark brown reddish in Thai specimens vs. dark brown in the holotype). Thailand is located within two significant biodiversity hotspots, Indo-Burma and Sundaland, and accommodates a high biodiversity of flora and fauna [33]; therefore, these differences are common among populations of the same species and can be attributed to genetic and bioclimatic factors.

*Microplitis manilae* looks similar to *M. aprilae* Austin and Dangerfield, 1993, *M. jamesi* Austin and Dangerfield, 1993, and *M. abrs* Austin and Dangerfield, 1993. However, *M*. *manilae* can be separated from *M*. *aprilae* and *M*. *jamesi* by (1) a strongly sculptured mesoscutum, (2) a more elongated T1, and (3) the presence of a median field on T2. *Microplitis manilae* is differentiated from *M*. *aprilae* by the margins of the mesoscutellum having an incomplete crenulate furrow, whereas *M. aprilae* is bordered laterally by a complete crenulate furrow, forming a distinct carinate margin. *Microplitis manilae* differs from *M. abrs* by (1) having a medial furrow on the mesoscutum, (2) a strongly sculptured mesoscutum and mesoscutellum, and (3) a laterally rugose-punctate T1 and an apical surface with smooth convex swelling. It is worth noting that specimens from Queensland were only tentatively assigned to *M. manilae* as they fall outside the level of variation described, but they are not distinctive enough to be described as a separate species [23]. This is because in the original description of *M. manilae*, no variation in the color was mentioned since color was not an important distinguishing feature when it came to identifying the species. So, further studies are needed to determine if the specimens warranted species-level taxonomic recognition. This Queensland material (now deposited in the ANIC (Australian National Insect Collection, Canberra, Australia)) differs from the Thai specimens in having a much smoother mesoscutum, a larger precoxal groove, and a T1 parallel to the apex. Several specimens from more southerly localities in Australia (Australian Capital Territory (ACT), Victoria, and Tasmania) differ even further from the type series in color, the sculpturing of the mesoscutum and/or mesoscutellum, and the shape of TI, and they probably represent several distinct endemic species. These differences suggest that they may have adapted to their local environment, and this adaptation has caused them to evolve differently from specimens in the type series. These specimens have not been described as each one is represented by a singleton, making it difficult to draw any conclusions about the species as a whole since the sample size is so small [34]. Furthermore, with the available data and from analyzing the molecular data along with host associations, we could verify whether the specimens are distinct species or not. This verification process would allow for a more accurate assessment of species diversity and provide a reliable baseline for future research.

### 4.2. Host-Parasitoid-Food Plant Associations

In biological research, names of species are essential to ensure comparable results when working with model organisms [35], and in agriculture, they are also required for biosecurity and quarantine concerns [36]. Notwithstanding, in some cases, species identification is not an easy task and deep taxonomic studies are needed. For instance, *M. manilae* might be misidentified as *M. similis* since the two species resemble each other, and both species attack *S. litura* (F.). Accordingly, only a detailed study based on a series of morphological characters and DNA barcoding would give support to the separation and validity of both species. As already mentioned, *Spodoptera* larvae are one of the main hosts for the *Microplitis* species, and because of their possible broad dietary tolerance, some species might migrate and disperse to new geographic regions. This might also promote the introduction and dispersion of exotic or new species of parasitoids in other regions. On the other hand, there are different species of *Spodoptera* used as hosts by some unidentified *Microplitis* species. Therefore, the diversity of this genus might be currently underestimated in those regions (e.g., Nearctic and Neotropical). In agriculture, species identification protocols based on molecular data represent powerful tools for the success of early detection programs or monitoring of species [36]. However, for some groups of insects, a lack of reference barcodes, errors in databases, a scarcity of voucher specimens, and the presence of cryptic species represent strong limitations. As an example, a series of recently known Microgastrinae species was described based on morphological characters [18], but because their sampling seems to be seasonally restricted, the generation of barcodes for further studies on their biology and phylogeny represented a challenge.

The reliability of host data depends on several factors such as (a) sampling method, (b) host identification, (c) sample size, (d) host age and condition, (e) geographic location, and (f) host-parasitoid interaction [37], and until illustrated evidence, even multimedia evidence with detailed biology data, is published, it will remain questionable whether the host(s) are actual or potential. Above, information about the food plants, lifestyle, morphology, and molecular data for all 15 *Microplitis* species attacking *Spodoptera* species are summarized (Table 1). There is a dearth of molecular studies, which will require special attention in the future.

### 4.3. Ecological Niche Modelling

The MaxEnt model showed that the highly suitable areas were predominantly located in tropical and subtropical countries. The climate of these regions is very hot and humid. The average temperature during every month is above 18 °C, and throughout the year, the temperature remains relatively constant (warm) and the sunlight is intense. There is no winter season, these areas are frost-free, and the annual rainfall is large and exceeds the annual evaporation. The findings suggested that the model’s prediction performance regarding the distribution of *M. manilae* was notably good, with a considerably high level of reliability.

In this research, five bioclimatic variables were important in limiting *M. manilae* distribution: BIO13, BIO12 (precipitation), BIO1, BIO4, and BIO10 (temperature). Precipitation and temperature jointly constrained the current distribution pattern of *M. manilae*. *Microplitis manilae* has a comparatively high tolerance to temperature and humidity and can develop effectively under different temperature conditions [12]. The ranges of BIO13 (precipitation during the wettest month) and BIO12 (annual precipitation) were relatively wide, which shows the adaptation of *M. manilae* to those environmental conditions. It has been reported that the parasitism rate, developmental duration, and longevity of *M. manilae* are inversely related to temperature and that the reproductive cycle, survival rate, and other biological parameters were significantly diverse under different temperatures and humidity [9,12]. Consequently, temperature and humidity play a key role in affecting the growth and development of *M. manilae* [9,12]. In this study, the suitable distribution range of BIO12 was 1428.1–3982.5 mm, and an elevation above 350.3 m was deemed unsuitable for the distribution of *M. manilae*. The temperature and humidity in tropical and subtropical areas are in line with the ideal environment for *M. manilae*. The majority of unsuitable distribution areas for *M. manilae* are predominantly situated in the temperate zone (Figure 5). Typically, the climate in these regions may be characterized by strong solar radiation, large/marked temperature differences between day and night, and low temperatures, which is not conducive to the survival of *M. manilae*.

In the future scenario of the 2070s, the total suitable area of *M. manilae* increased overall under different RCP combinations. However, incremental changes were not particularly noticeable. This may have been related to the wide ecological range of *M. manilae*, which was capable of adapting to a wide variety of environments. Specifically, *M. manilae* was found to be resilient to increasing temperatures and changes in precipitation, suggesting that it could thrive in a variety of climate scenarios. A potential consequence of climate warming is the expansion of suitable habitats for *M. manilae*, and it is anticipated that suitable habitats will shift towards higher latitudes and elevations in the future. This is because warmer temperatures can create more hospitable environments for *M. manilae*, allowing them to survive and reproduce better.

*Microplitis manilae* has been widely used as a biological control agent against *Spodoptera* larvae because it is one of their dominant natural enemies [9,10]. The topic of the large-scale and cost-effective rearing of *M. manilae* has sparked one of the most considerable debates in the field of biological control applications. *Microplitis manilae* has a relatively wide distribution area and is very dependent on the availability of its host species; consequently, the distribution of the host will affect the distribution of *M. manilae*. There are eight *Spodoptera* species for *M. manilae*: *S. cilium* Guenée, *S. depravata* (Butler), *S. exempta* (Walker), *S. exigua* (Hübner), *S. frugiperda* (J. E. Smith), *S. littoralis* (Boisduval), *S. litura* (Fabricius), and *S. mauritia* (Boisduval), which have caused significant economic losses (Table 1). According to the obtained distributions of hosts on the GBIF (Global Biodiversity Information Facility) website and the present data, the environmental conditions suitable for the above-mentioned lepidopteran hosts and *M. manilae* extensively overlap, and *M. manilae* could be used to target specific pests and reduce the need for chemical pesticides.

## 5. Conclusions

In the present study, we discuss the taxonomic and biological aspects of the economically important parasitoid wasp *M. manilae*. In addition, we construct ecological models (MaxEnt model and QGIS technology) which we hope will provide a reference framework for further monitoring programs or studies. The MaxEnt model suggests that *M. manilae* is well-adapted to a wide range of tropical and subtropical climates. Five bioclimatic variables appeared to be particularly important for predicting the species distribution: BIO13 (precipitation during the wettest month), BIO12 (annual precipitation), BIO1 (annual mean temperature), BIO4 (temperature seasonality), and BIO10 (mean temperature during the warmest quarter). A discernible trend of further expansion was observed in the distribution range of *M. manilae* within highly suitable areas. This study is expected to serve as a valuable resource for advancing the current understanding of the environmental factors that influence the distribution of *M. manilae*, which implies a significant potential for using this parasitoid wasp to control *Spodoptera* larvae because most of the areas occupied by this pest belong to temperate and subtropical regions in Southeast Asia.

## Figures and Tables

**Figure 1 insects-14-00338-f001:**
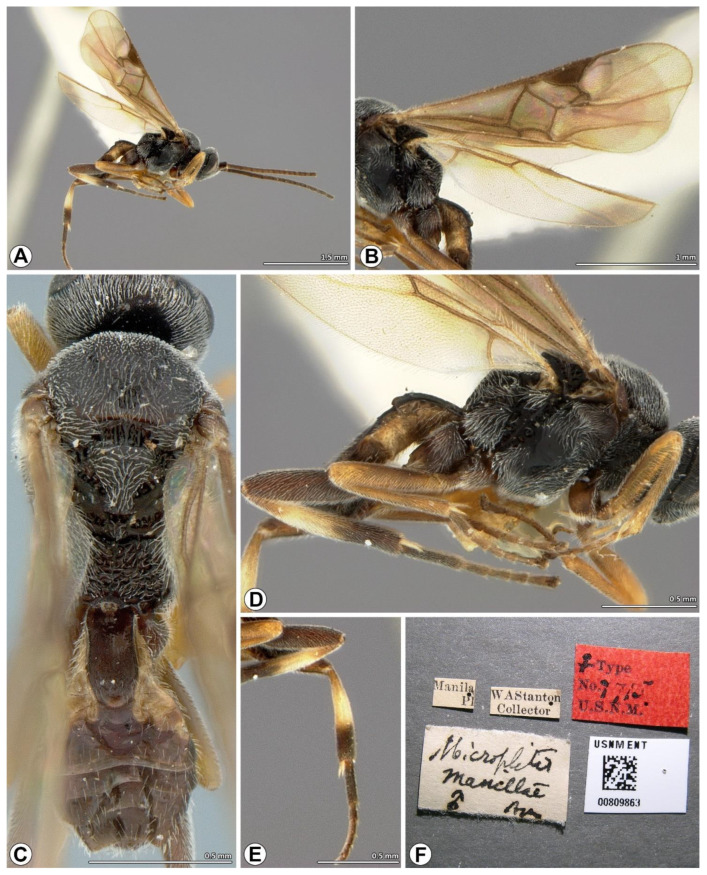
*Microplitis manilae* Ashmead, 1904, holotype female from the Philippines (USNM). (**A**) Habitus, lateral view; (**B**) wings; (**C**) head, mesosoma, and metasoma, dorsal view; (**D**) mesosoma and metasoma, lateral view; (**E**) hind leg; (**F**) holotype labels.

**Figure 2 insects-14-00338-f002:**
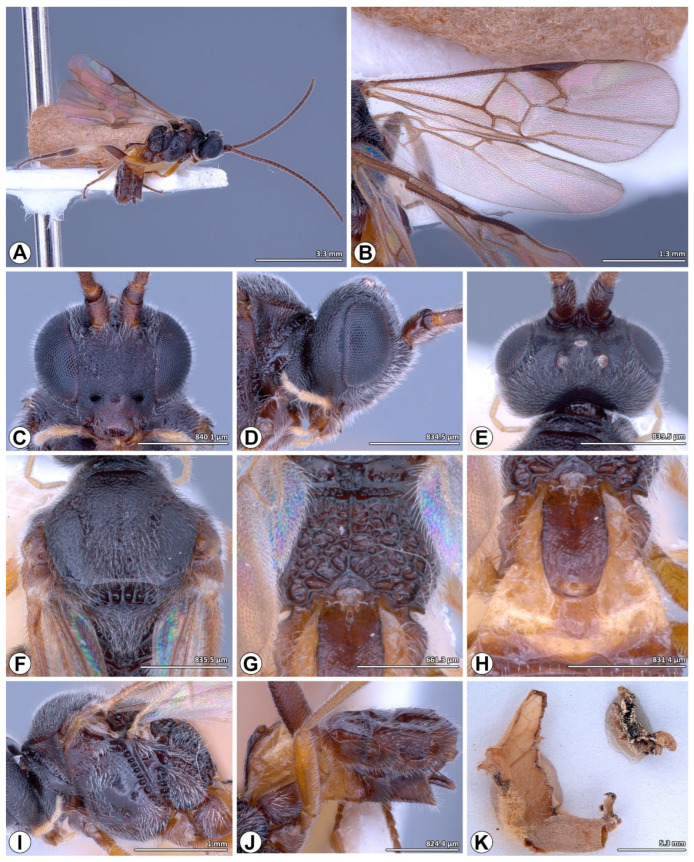
*Microplitis manilae* Ashmead, 1904, non-type female from Thailand (CUMZ). (**A**) Habitus, lateral view; (**B**) wings; (**C**–**E**) head; (**C**) frontal view; (**D**) lateral view; (**E**) dorsal view; (**F**) mesoscutum and mesoscutellum, dorsal view; (**G**) metanotum and propodeum, dorsal view; (**H**) T1–T2, dorsal view; (**I**) mesosoma, lateral view; (**J**) metasoma, lateral view; (**K**) wasp cocoon (left) and larvae of *Spodoptera litura* (**F**) (upper right).

**Figure 3 insects-14-00338-f003:**
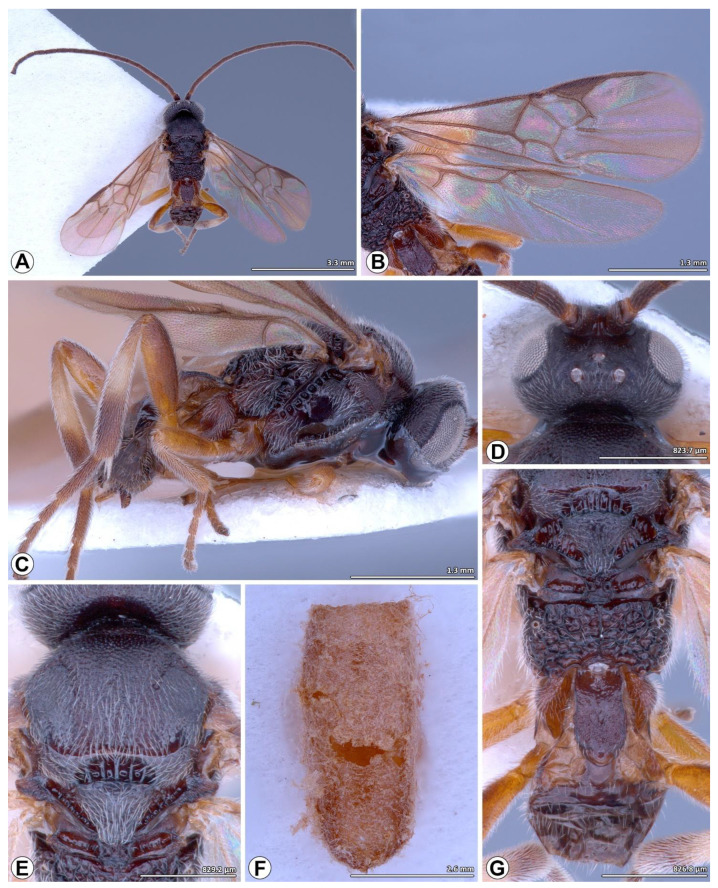
*Microplitis manilae* Ashmead, 1904, non-type male from Thailand (CUMZ). (**A**,**C**) Habitus; (**A**) dorsal view; (**C**) lateral view; (**B**) wings; (**D**) head, dorsal view; (**E**) mesoscutum and mesoscutellum, dorsal view; (**F**) wasp cocoon; (**G**) mesoscutellum, metanotum, propodeum, and metasoma dorsal view.

**Figure 4 insects-14-00338-f004:**
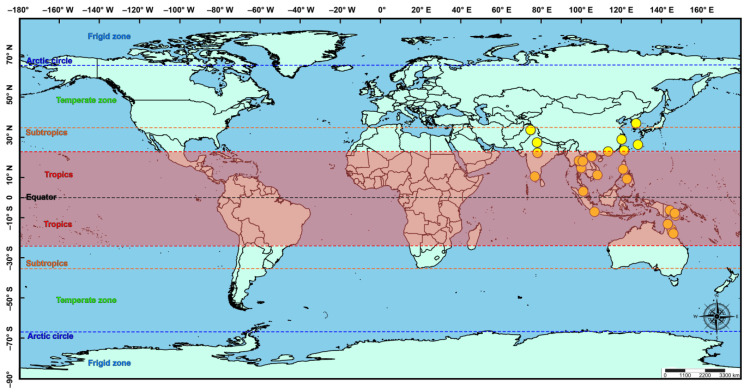
Current geographical distribution of *Microplitis manilae* in a global context, displayed in yellow points.

**Figure 5 insects-14-00338-f005:**
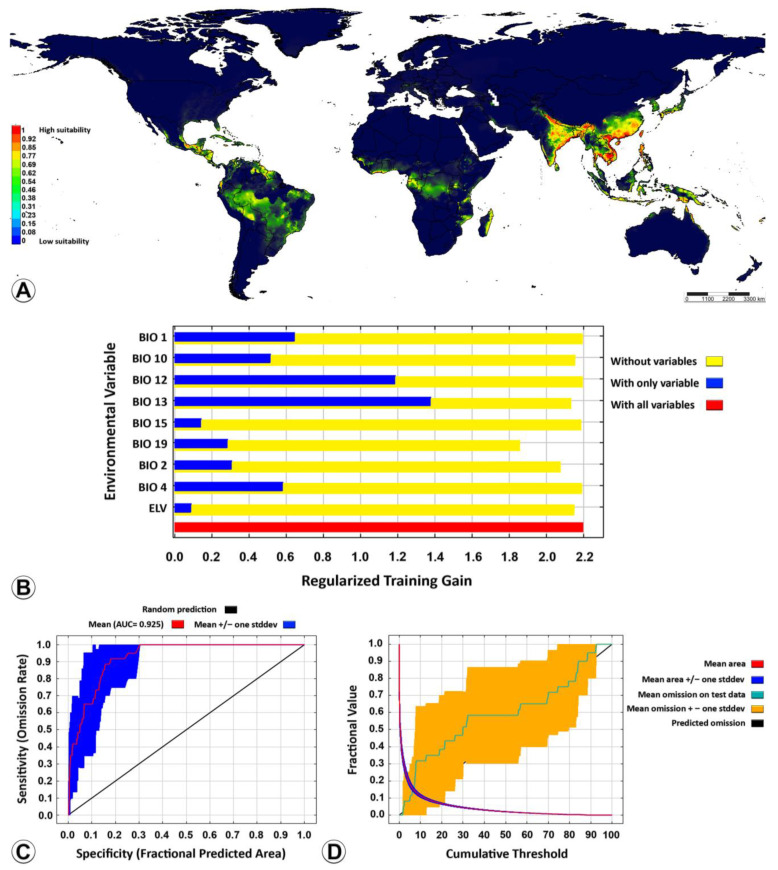
MaxEnt outputs of *Microplitis manilae* under its current distribution. (**A**) Current suitable distribution in a global context; (**B**) importance of bioclimatic variables to *M. manilae* by Jackknife test; (**C**) ROC curve of potential distribution prediction; (**D**) curve of omission and predicted area. High-suitability areas have a probability of 1–0.77; medium-suitability areas have a probability of 0.77–0.46; low-suitability areas have a probability of 0.46–0.23; unsuitable areas have a probability of 0.23–0. AUC = area under the curve; MaxEnt = maximum entropy modelling; ROC = receiver operating characteristic curve.

**Figure 6 insects-14-00338-f006:**
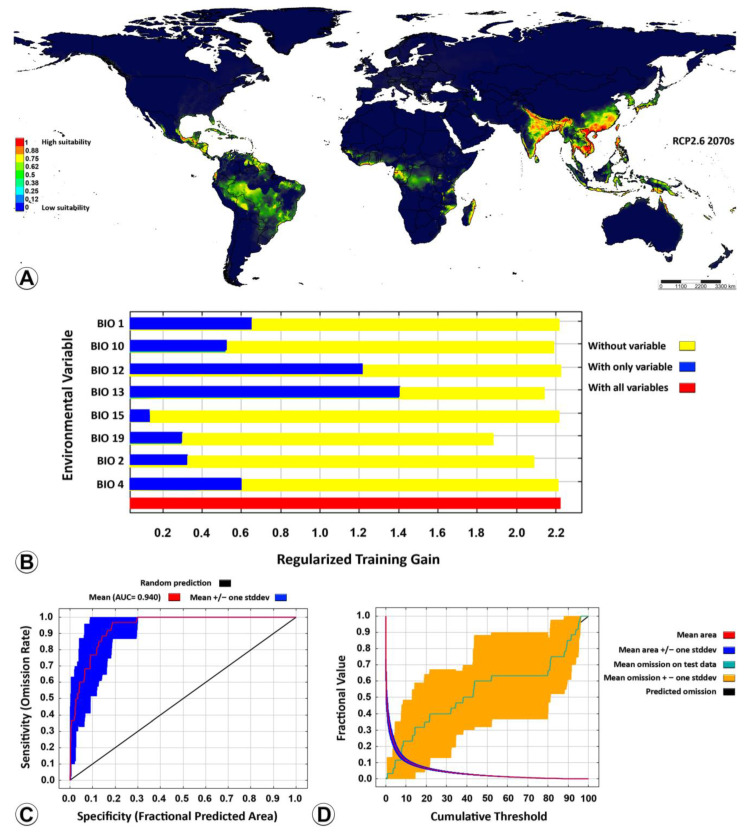
Potential distribution of *Microplitis manilae* under the RCP2.6 climate change scenario in the 2070s. (**A**) Future suitable distribution under the RCP2.6 scenario in a global context; (**B**) importance of bioclimatic variables to *M. manilae* by Jackknife test; (**C**) ROC curve of potential distribution prediction; (**D**) curve of omission and predicted area. High-suitability areas has a probability of 1–0.75; medium-suitability areas have a probability of 0.75–0.5; low-suitability areas have a probability of 0.5–0.25; unsuitable areas have a probability of 0.25–0. ROC = receiver operating characteristic curve; RCP = representative concentration pathways.

**Figure 7 insects-14-00338-f007:**
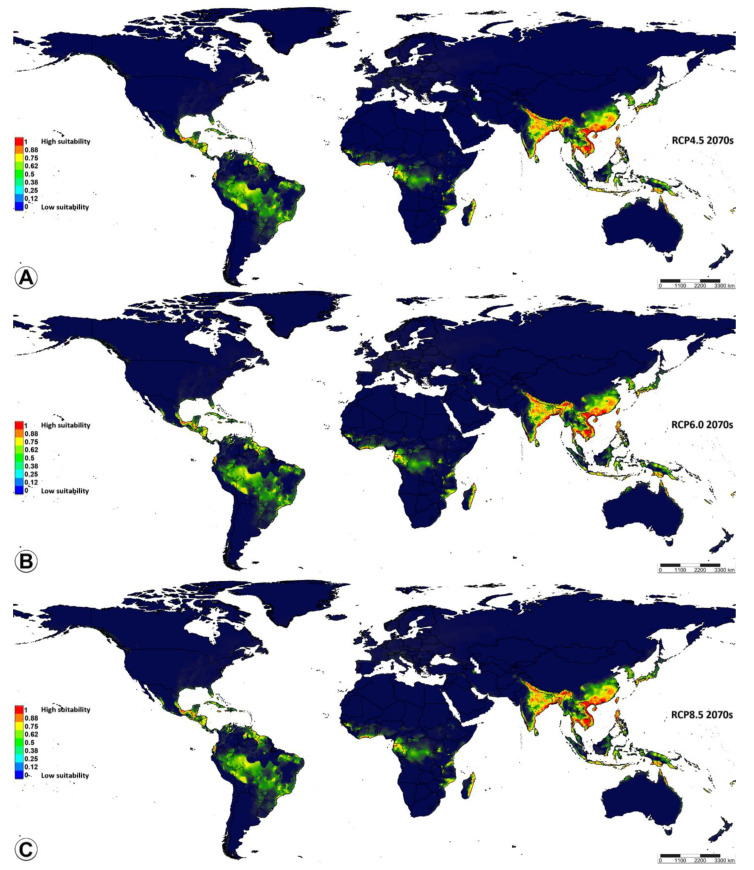
Potential distribution of *Microplitis manilae* under three climate change scenarios in the period of 2070s. (**A**–**C**) Future suitable distribution in a global context under three scenarios: (**A**) RCP4.5; (**B**) RCP6.0; (**C**) RCP8.5. High-suitability areas have a probability of 1–0.75; medium-suitability areas have a probability of 0.75–0.5; low-suitability areas have a probability of 0.5–0.25; unsuitable areas have a probability of 0.25–0. RCP = representative concentration pathways.

**Figure 8 insects-14-00338-f008:**
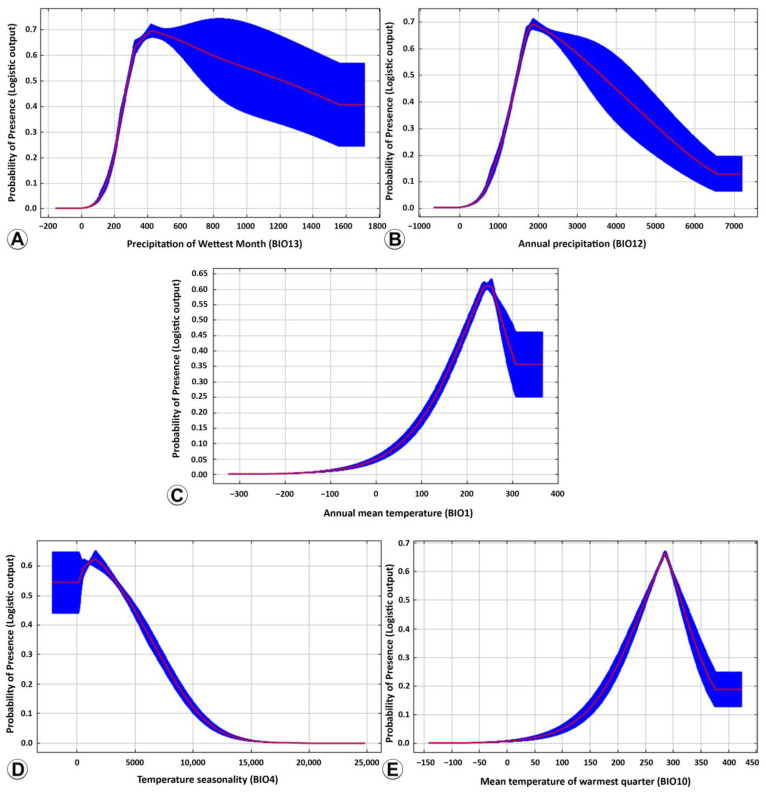
Response curves between bioclimatic variables and predicted suitability. (**A**) Precipitation of the wettest month (BIO13); (**B**) annual precipitation (BIO12); (**C**) annual mean temperature (BIO1); (**D**) temperature seasonality (BIO4); (**E**) mean temperature during the warmest quarter (BIO10).

**Table 1 insects-14-00338-t001:** Bioclimatic variables are often used in species distribution modelling and related ecological modelling techniques. The variables are derived from minimum, maximum, and mean temperature and mean precipitation values and are related to the distribution of *Microplitis manilae*.

Code	Bioclimatic Variables	Units	Being Used for Modelling
BIO1	Annual mean temperature	°C	Yes
BIO2	Mean diurnal range (mean of monthly temp (max temp–min temp))	°C	Yes
BIO3	Isothermality (BIO2/BIO7) × 100	°C	No
BIO4	Temperature seasonality	SD × 100	Yes
BIO5	Max temperature during the warmest month	°C	No
BIO6	Min temperature during the coldest month	°C	No
BIO7	Annual temperature range (BIO5–BIO6)	°C	No
BIO8	Mean temperature during wettest quarter	°C	No
BIO9	Mean temperature during driest quarter	°C	No
BIO10	Mean temperature during warmest quarter	°C	Yes
BIO11	Mean temperature during coldest quarter	°C	No
BIO12	Annual precipitation	mm	Yes
BIO13	Precipitation during the wettest month	mm	Yes
BIO14	Precipitation during the driest month	mm	No
BIO15	Precipitation seasonality	CV	Yes
BIO16	Precipitation during the wettest quarter	mm	No
BIO17	Precipitation during the driest quarter	mm	No
BIO18	Precipitation during the warmest quarter	mm	No
BIO19	Precipitation during the coldest quarter	mm	Yes
ELV	Elevation	m	Yes

Codes represent annual trends (e.g., mean annual temperature, annual precipitation), seasonality (e.g., annual range in temperature and precipitation), and extreme or limiting environmental factors (e.g., the temperature during the coldest and warmest month and precipitation during the wet and dry quarters). A quarter is a three-month period (1/4 of the year). CV = coefficient of variation; SD = standard deviation.

**Table 2 insects-14-00338-t002:** List of all *Microplitis* species (Hymenoptera: Braconidae) recorded as attacking *Spodoptera* spp. (Lepidoptera: Noctuidae). Larvae: S–wasp larvae solitary; S?–wasp larvae suspected to be solitary but no conclusive evidence; G–wasp larvae gregarious. ?–unknown. In cases where a species has both solitary and gregarious larvae, the most common occurrence is indicated first. MOR, DNA, BIO: degree of species support by morphological (MOR), molecular (DNA), and biological (BIO) data. “+” strong support, “–” no support, “P” partial support, “?” unknown.

Species	Author	Host	Host Plant	Cocoon	Larvae	MOR	MOL	BIO
*Microplitis* sp.	–	*S. depravata* (Butler)	?	?	?	–	–	P
*M. abrs*	Austin and Dangerfield, 1993	*S. litura* (Fabricius)	?	?	S	+	–	+
		*S. exigua* (Hübner)?						
*M. ajmerensis*	Rao and Kurian, 1950	*S. exigua* (Hübner)	?	?	S?	+	–	P
*M. albotibialis*	Telenga, 1955	*S. exigua* (Hübner)	?	?	S?	+	–	P
*M. bicoloratus*	Chen, 2004	*S. litura* (Fabricius)	*Arachis hypogaea* L.	?	S	+	–	+
*M. demolitor*	Wilkinson, 1934	*S. frugiperda* (J. E. Smith)	?	?	S	+	+	+
		*S. littoralis* (Boisduval)	?					
		*S. litura* (Fabricius)	?					
*M. fulvicornis*	(Wesmael, 1837)	*S. exigua* (Hübner)	*Beta vulgaris* L.	White grayish	S	+	–	+
*M. leucaniae*	Xu and He, 2002	*S. litura* (Fabricius)	?	?	S?	+	–	P
*M. manilae*	Ashmead, 1904	*S. exempta* (Walker)	?	Light brown	S	+	P	+
		*S. exigua* (Hübner)	?					
		*S. littoralis* (Boisduval)	?					
		*S. litura* (Fabricius)	*Nicotiana tabacum* L. *Glycine max* (L.) *Brassica oleracea* L. (var. *italica*, *capitata*, *botrytis*)					
		*S. frugiperda* (J. E. Smith)	*Zea mays* L.					
*M. pallidipes*	Szépligeti, 1902	*S. litura* (Fabricius)	?	?	S	+	–	+
		*S. exigua* (Hübner)	?					
*M. prodeniae*	Rao and Kurian, 1950	*S. litura* (Fabricius)	*Amaranthus* sp. *Nicotiana tabacum* L.	Light brown	S	+	+	+
*M. rufiventris*	Kokujev, 1914	*S. cilium* Guenée	?	Tan	S, G	+	+	+
		*S. exigua* (Hübner)	*Medicago sativa* L. *Zea mays* L.					
		*S. littoralis* (Boisduval)	*Gossypium hirsutum* L.					
		*S. frugiperda* (J. E. Smith)	*Glycine max* (L.)					
*M. similis*	Lyle, 1921	*S. litura* (Fabricius)	*Glycine max* (L.)	?	S, G	+	+	+
*M. spectabilis*	(Haliday, 1834)	*S. exigua* (Hübner)	?	?	S?	+	–	P
*M. spodopterae*	Rao and Kurian, 1950	*S. mauritia* (Boisduval)	*Trigonella foenum-graecum* L.	Brown	S	+	–	+
*M. tuberculifer*	(Wesmael, 1837)	*S. exigua* (Hübner)	*Zea mays* L.	?	S	+	P	+
		*S. litura* (Fabricius)	?					

**Table 3 insects-14-00338-t003:** Percent contribution and the permutation importance of bioclimatic variables affecting the distribution of *Microplitis manilae*.

Code	Percent Contribution/%	Permutation Importance/%
**At present time and under the climate change scenario RCP2.6 2070s**		
BIO13	67.4	43.5
BIO19	18.6	21.5
BIO2	5.3	5.6
BIO10	2.5	20.9
ALT	2.4	5.1
BIO4	1.9	0.4
BIO1	1	1.5
BIO15	0.5	1.2
BIO12	0.4	0.3
**Under climate change scenarios RCP4.5, RCP6.0, and RCP8.5 in the period of the 2070s**		
BIO13	66.9	60.5
BIO19	20.1	22.2
BIO2	6.3	4.3
BIO10	2.9	4.7
BIO4	1.9	5.3
BIO1	0.9	2.4
BIO12	0.9	0.1
BIO15	0.3	0.6

**Table 4 insects-14-00338-t004:** Overlap of predicted suitable areas for *Microplitis manilae* under current and future (RCP2.6 2070s, RCP4.5 2070s, RCP6.0 2070s, and RCP8.5 2070s) climatic conditions.

Niche Overlap	Current	RCP8.5 2070s	RCP6.0 2070s	RCP4.5 2070s	RCP2.6 2070s
**Current**	1				
**RCP8.5 2070s**	0.776919293	1			
**RCP6.0 2070s**	0.812357129	0.881662813	1		
**RCP4.5 2070s**	0.812198045	0.885771393	0.910782646	1	
**RCP2.6 2070s**	0.842805348	0.85744628	0.8886515	0.895545884	1

**Table 5 insects-14-00338-t005:** Suitable range of bioclimatic variables for the potential distribution of *Microplitis manilae*.

Bioclimatic Variables	Suitable Range	Optimum Value
BIO13 (mm)	269.2–763.4	435.8
BIO19 (mm)	175.1–704.8	342.3
BIO2 (°C)	64.3–107.4	80.6
BIO10 (°C)	198.4–326.2	292.4
ALT (m)	−57.0–350.3	−57.1
BIO4 (SD × 100)	132–4855.6	2240.7
BIO1 (°C)	197.3–286.4	246.7
BIO15 (CV)	59.4–160.1	83.5
BIO12 (mm)	1428.1–3982.5	1876.3

## Data Availability

The data presented in this study are available from the corresponding author upon reasonable request.

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
