# Peer review of "Microplitis manilae Ashmead (Hymenoptera: Braconidae): Biology, Systematics, and Response to Climate Change through Ecological Niche Modelling"

_insects, 2023, doi:10.3390/insects14040338_

Round 1

Reviewer 1 Report

 Summary

The manuscript describes the biology and distribution of the parasitoid wasp Microplitis manilae, a biocontrol agent that exclusively attacks caterpillars. In particular, the genus Spodoptera which includes armyworms, is a major pest species in the western hemisphere; however, little is known concerning the associations between the wasp and the armyworm. The paper comprises three parts: a redescription of Microplitis manilae based on the holotype, a narrative describing the host-parasitoid-food plant associations of the genus Microplitis and Spodoptera, and lastly, current and future distribution modeling of M. manilae based on climate data. The authors conclude that M. manilae is adapted to a wide range of climates and that its suitable range is expected to shift to higher latitudes and altitudes in the future due to climate changes.

General comments

This manuscript includes several objectives, each of which is interesting on its own. The manuscript could benefit from a little restructuring to inform the reader which objective the authors are describing in the methods and results. For instance, the numbering system could follow objectives 1-3 and be labeled as such in all sections, not just the discussion. Overall, the introduction and discussion are informative but could use a little more foundation from the published literature. The methods lack some detail that would allow for reproducibility, and the figures have limited accessibility to all readers (see line comments below).

Specific comments

Lines 48-49: There is no end parenthesis for the example of macrolepidopterans given.

Lines 79-82: Please provide some example papers.

Line 105: It is unclear what “Thai specimens” means. Are you referring to the wasp or armyworm, or both? Is this one species or several? Additionally, where was the collection made, and in what year were the specimen collected?

Section 2.2: Does this section refer to the same specimen as in line 105? If so, maintain the nomenclature of calling these “Thai specimen”. If not, then clarification is needed.

Lines 125-126: Did you sample three lines of each of the three varieties of B. orleracea for nine different types of plants? Please clarify this statement, it can be read a few different ways.

Lines 126-127: What was the location that the caterpillars were caught from?

Line 147: Please provide a reference for “the Buffer analysis method”.

Line 157: WorldClim provides elevation data from the SRTM data, not altitude. If this is the data used, the term elevation (height in reference to the mean sea level) is correct, whereas the altitude (height above the surface of the ground) is less appropriate. Although these terms can be interchangeable, elevation directly describes the data.

Lines 182-184: Are these four levels divided equally within the 0-1 range? In other words, is high suitability within the 0.75 – 1 range and the other three each quartile range below? Please clarify.

Lines 184-193: What models and steps to penalize complexity were included? For instance, linear, quadratic, and hinge models with penalty range 1-4 at a step value of 1.

Lines 188-189: Clarify what is meant by, ”Only the optimal parameters were used to simulate and predict the suitable 188 habitat of M. manilae in different periods”. Is this in reference to MaxEnt automatically removing climate variables that do not add information to the model, or were only non-correlated variables included in the model after you removed highly correlated ones? If you did remove highly correlated variables, then please state how you did so. If you did not, state why this was not done, as it is commonplace to do so when creating species distribution models (SDMs).

Line 336: Which model was selected by MaxEnt?

Line 352: What was the OR value?

Lines 383-384: This statement contradicts the previous statement that “all the predicted suitable areas (high, medium, and low) showed 378 a trend of expansion”. Please clarify how there is both expansion and a decrease. Additionally, how great is the increase or decrease? It is difficult to see these changes in the figures, so adding an effect size here would be informative.

Fig 5—7: It appears that area was removed from the analysis or covered post-model building. However, these methods were not mentioned, nor is the black area noted in the figures. Please provide how and why this was done. For instance, if you used a similarity index, such as MESS, and revealed areas above a threshold.

Fig 5—7: A color scheme readable to those with red-green blindness would make these maps more accessible. This pertains to both the maps as well as graphs.

Lines 557-559: There is a distinct lack of comparisons to other SDM literature within the discussion. Here is one example of a location to inform the reader if these same climate predictions have been noted with other species.

Author Response

Dear Reviewer,

Thank you very much for reviewing our MS and gave valuable comments. We did all the corrections and clarify some unclear text. 

Best regards,

The authors

Reviewer 2 Report

The manuscript is a nice contribution for modeling a parasitoid using bioclimatic variables and QGIS tools via Maxent. The authors have described the morphological identification of species using High tech microphotography. However I have some suggestions for the improvement of manuscript attached in a separate file.

Author Response

Dear Reviewer,

Thank you very much for reviewing our MS. We did all the corrections as suggested. 

Thank you,

The authors

Round 2

Reviewer 1 Report

Thank you for your edits and comments concerning the first review. The manuscript is much stronger. The methods could be more easily reproduced. Additionally, the results are much more detailed. I particularly appreciate the addition of lines 429-441, which reveal a clearer picture of the distribution changes. I have no further individual line comments. 

Author Response

Dear the reviewer,

Thank you very much for your kind comments. We are appreciated it.

Best regards,

Buntika and Mostafa